# Givens Coordinate Descent Methods for Rotation Matrix Learning in Trainable Embedding Indexes

**Yunjiang Jiang***, **Han Zhang, Yiming Qiu, Yun Xiao, Bo Long, Wen-Yun Yang***
JD.com, Mountain View, CA, United States & Beijing, China
{yunjiangster,wenyun.yang}@gmail.com
{zhanghan33,qiuyiming3,xiaoyun1,bo.long}@jd.com

## Abstract

Product quantization (PQ) coupled with a space rotation, is widely used in modern approximate nearest neighbor (ANN) search systems to significantly compress the disk storage for embeddings and speed up the inner product computation. Existing rotation learning methods, however, minimize quantization distortion for fixed embeddings, which are not applicable to an end-to-end training scenario where embeddings are updated constantly. In this paper, based on geometric intuitions from Lie group theory, in particular the special orthogonal group $SO(n)$, we propose a family of block Givens coordinate descent algorithms to learn rotation matrix that are provably convergent on any convex objectives. Compared to the state-of-the-art SVD method, the Givens algorithms are much more parallelizable, reducing runtime by orders of magnitude on modern GPUs, and converge more stably according to experimental studies. They further improve upon vanilla product quantization significantly in an end-to-end training scenario.

## 1 Introduction

Search index is the core technology to enable fast information retrieval in various modern computational systems, such as web search, e-commerce search, recommendation and advertising in the past few decades. As a traditional type of search index, inverted index (Dean, 2009), which maps terms to documents in order to retrieve documents by term matching, have been the mainstream type of search index for decades, thanks to its efficiency and straightforward interpretability. Recently, with the advent of deep learning era, embedding indexes coupled with approximate nearest neighbor (ANN) search algorithms, have established as a promising alternative to search index (Zhang et al., 2020) and recommendation index (Covington et al., 2016; Huang et al., 2020; Li et al., 2019), in part due to its learnable representations and efficient ANN algorithms.

The main idea of embedding indexes is to encode users (or queries) and items in a latent vector space, and represent their semantic proximity in terms of inner product or cosine similarity. Embedding indexes enjoy a few appealing characteristics: a) the embeddings can be learned to optimize downstream retrieval task of interests, and b) items can be efficiently retrieved within tens of milliseconds. The latter leverages decades of algorithmic innovations, including a) maximum inner product search (MIPS) or approximate nearest neighbors (ANN), such as locality sensitive hashing (LSH) (Datar et al., 2004), hierarchical navigable small world graphs (HNSW) (Malkov & Yashunin, 2020), space indexing by trees (Bentley, 1975; Dasgupta & Freund, 2008; Muja & Lowe, 2014; Bernhardsson, 2018) or graphs (Harwood & Drummond, 2016; Iwasaki, 2015), and b) state-of-the-art product quantization (PQ) based approaches (Jegou et al., 2010; Johnson et al., 2019; Guo et al., 2020; Wu et al., 2017). In particular, PQ based indexes and its variants have regularly claimed top spots on public benchmarks such as GIST1M, SIFT1B (Jegou et al., 2010) and DEEP10M (Yandex & Lempitsky, 2016). Moreover, along with the open-sourced libraries Faiss (Johnson et al., 2019) and ScaNN (Guo et al., 2020), PQ based embedding indexes are widely adopted in many industrial systems (Huang et al., 2020; Zhang et al., 2020).

---

*Corresponding author

## 1.1 PRODUCT QUANTIZATION (PQ) BASED INDEXES

Since Jegou et al. (2010) first introduced PQ and Asymmetric Distance Computation (ADC) from signal processing to ANN search problem, there have been multiple lines of research focused on improving PQ based indexes. We briefly summarize the main directions below.

**Coarse Quantization**, also referred to as inverted file (IVF), is introduced in the original work of Jegou et al. (2010) to first learn a full vector quantization (referred to as coarse quantization) by k-means clustering and then perform product quantization over the residual of the coarse quantization. This enables non-exhaustive search of only a subsets of the clusters, which allows ANN to retrieve billions of embeddings in tens of milliseconds. More work has been done in this direction, including Inverted Multi-Index (IMI) (Lempitsky, 2012);

**Implementation Optimization** efforts have mainly been spent on the computation of ADC, including using Hamming distance for fast pruning (Douze et al., 2016), an efficient GPU implementation of ADC lookup (Johnson et al., 2019), and SIMD-based computation for lower bounds of ADC (André et al., 2015). Our proposed method is fully compatible with all these implementation optimizations, since we are solely focused on the rotation matrix learning algorithms.

**Rotation Matrix Learning** is introduced to reduce the dependencies between PQ subspaces, since PQ works best when the different subspaces are statistically independent, which may not be true in practice. As early as 2013, Optimized PQ (OPQ) (Ge et al., 2013), ITQ (Gong et al., 2013) and Cartesian k-means (Norouzi & Fleet, 2013) all propose the idea of alternating between learning (product) vector quantization and that of the rotation matrix; the latter can be formulated as the classic Orthogonal Procrustes problem (Schönemann, 1966) that has a closed form solution in terms of singular value decomposition (SVD), which is also widely used in CV (Levinson et al., 2020).

**Cayley Transform** (Cayley, 1846) recently inspires researcher to develop end-to-end algorithm to learn rotation matrix in various neural networks, especially unitary RNN (Helfrich et al., 2018; Casado & Martínez-Rubio, 2019) and PQ indexes (Wu et al., 2017). Formally, Cayley transform parameterizes a $d \times d$ rotation matrix $R$ by $R = (I - A)(I + A)^{-1}$, where A is a skew-symmetric matrix (*i.e.*, $A = -A^\top$). Note that the above parameterization of $R$ is differentiable w.r.t. the $\binom{d}{2}$ parameters of $A$. Thus, it allows for end-to-end training of rotation matrix $R$, as long as the gradient of $R$ can be obtained. However, as we will discuss in Section 2.4, these Cayley transform based methods are not easily parallelizable to modern GPUs thus inefficient compared to our proposed method. Another drawback of Cayley transforms is numerical instability near orthogonal matrices with -1 eigenvalues. This has been partially addressed in (Lezcano-Casado & Martínez-Rubio, 2019) using approximate matrix exponential methods. However the latter incurs higher computation cost per step, due to the higher order rational functions involved. The question of minimal Euclidean dimension to embed $SO(n)$ has been shown in (Zhou et al., 2019) to be of order $O(n^2)$. By contrast, our proposed GCD methods only require $O(n)$ dimension of **local** parameterization.

## 1.2 TRAINABLE INDEXES

Despite its various advantages, a notable drawback of embedding indexes is the separation between model training and index building, which results in extra index building time and reduced retrieval accuracy. Typically, for a large industrial dataset with hundreds of millions of examples, it takes hours to build the index, and recall rates drop by at least 10% (Zhang et al., 2021). This spurred a new trend of replacing embedding indexes with jointly learnable structural indexes. Researchers argue that product quantization based embedding indexes are suboptimal, as it can not be learned jointly with a retrieval model. To overcome this drawback, researchers propose a few learnable index structures, such as tree-based ones (Zhu et al., 2018; 2019), a K-D matrix one (Gao et al., 2020) and an improved tree-based one (Zhuo et al., 2020). Though these alternative approaches have also shown improved performance, they often require highly specialized approximate training techniques, whose complexities unfortunately hinder their wide adoptions.

On the other hand, learning PQ based indexes and retrieval model in an end-to-end scenario is also challenging, in large part due to the presence of non-differential operators in PQ, e.g., $\arg\min$. Recently, however, Zhang et al. (2021) proposes an end-to-end training method which tackles the non-differentiability problem by leveraging gradient straight-through estimator (Bengio et al., 2013), and it shows improved performance over the separately trained embedding indexes. However, learning

a rotation matrix in an end-to-end training scenario remains unaddressed. As alluded to earlier, all existing methods of learning rotation matrix are based on alternative minimization procedure with an expensive non-parallelizable SVD step in each iteration. Thus it is not applicable to various back-propagation based techniques in neural network learning.

Note that the orthonormality constraint on the rotation matrix is critical to ensure pairwise distances are preserved between query and item embeddings. In end to end training, however, one could relax this condition by using an L2 regularization loss of the form $\|XX^\top - I\|_2^2$. We have not explored this latter approach here, since we aim for applicability of the method even for standalone PQ.

### 1.3 OUR CONTRIBUTIONS

In this paper, we overcome the last hurdle to fully enable end-to-end training of PQ based embedding index with retrieval models, by leveraging mathematical studies of decomposition of orthogonal group by Hurwitz (1963) as early as 1963. Based on intuitions about maximal tori in compact Lie groups (Hall, 2015), we propose an efficient Givens coordinate gradient descent algorithm to iteratively learn the rotation matrix – starting from an identity matrix and applying a set of maximally independent (mutually commuting) Givens block rotations at each iteration. We open sourced the proposed algorithm, which can be easily integrated with standard neural network training algorithms, such as Adagrad and Adam.

Our contribution can be summarized as follows.

- Methodologically, we change the landscape of learning rotation matrix in approximate nearest neighbor (ANN) search from SVD based to iterative Givens rotation based, in order to be applicable to end-to-end neural network training.
- Algorithmically, we propose a family of Givens coordinate block descent algorithms with complexity analysis and convergence proof of the least effective variant, **GCD-R**.
- Empirically, we prove that for fixed embeddings the proposed algorithm is able to converge similarly as the existing rotation matrix learning algorithms, and for end-to-end training the proposed algorithm is able to learn the rotation matrix more effectively.

## 2 METHOD

### 2.1 REVISIT TRAINABLE PQ INDEX

Figure 1 illustrates an overview of an embedding indexing layer inside a typical deep retrieval two-tower model. Formally, the indexing layer defines a *full quantization function* $\mathcal{T} : \mathbb{R}^{m \times n} \to \mathbb{R}^{m \times n}$ that maps a batch of $m$ input embedding $X$ in $n$-dimensional space to output embeddings $\mathcal{T}(X)$, which can be decomposed into two functions: a *product quantization* function $\phi : \mathbb{R}^{m \times n} \to \mathbb{R}^{m \times n}$ and a *rotation* function with an orthonormal matrix $R$. The indexing layer multiplies the input embedding $X$ by a rotation matrix $R$ and the product quantized embedding $\phi(XR)$ by its inverse matrix $R^{-1}$, which is equal to its transpose $R^\top$ for orthonormal rotation matrix. Formally,

$$\mathcal{T}(X) = \phi(XR)R^\top.$$

We omit the definition of product quantization function $\phi$ here, since it is not the focus of this paper. Interested readers can find the full definitions in (Jegou et al., 2010) or (Zhang et al., 2021).

The final loss function includes two parts, a retrieval loss and a quantization distortion loss, formally,

$$\mathcal{L}(X) = \mathcal{L}_{ret}(\mathcal{T}(X)) + (1/m)\|XR - \phi(XR)\|^2 \tag{1}$$

where $\mathcal{L}_{ret}$ denotes any retrieval loss – typically by softmax cross entropy loss or hinge loss (Covington et al., 2016; Zhang et al., 2020; Huang et al., 2020). Note that straight-foward optimization of $\mathcal{L}$ is intractable, since the quantization function $\phi$ is not differentiable, which, however, can still be tackled by gradient straight-through estimator (Bengio et al., 2013). Then the only remaining unaddressed problem is how to iteratively learn the rotation matrix $R$, since existing methods only solve the case where the input vectors $X$ are fixed (Ge et al., 2013; Jegou et al., 2010). We will present our approach in the following sections.

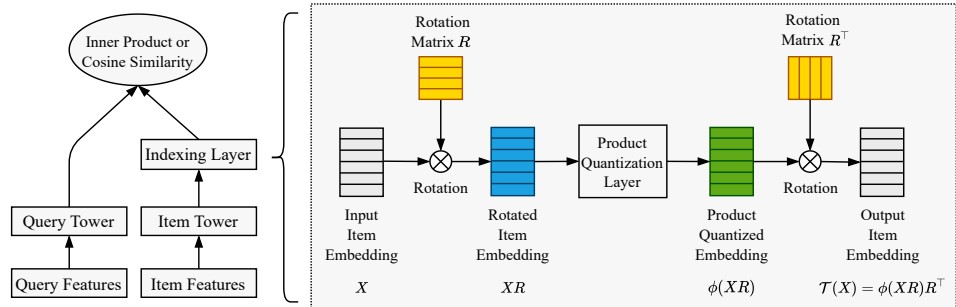

Figure 1: Illustration of an end-to-end trainable two-tower retrieval model. The indexing layer located at the top of item tower is composed of a rotation matrix and a product quantization layer.

## 2.2 GIVENS ROTATION

Before we present the algorithm of learning a rotation matrix, let us first present a few mathematical preliminaries that serve as foundations to our algorithm. For simplicity of exposition, the embedding dimension $n = 2m$ will always be an even number.

**Definition 1.** *Let $\mathbb{R}^{n \times n}$ denote the set of $n \times n$ matrices. An orthogonal group $O(n) = \{A \in \mathbb{R}^{n \times n} : AA^\top = \mathcal{I}_n\}$ is the group of all distance-preserving transformations of the Euclidean space $\mathbb{R}^n$. The special orthogonal group, denoted $SO(n)$, consists of all orthogonal matrices with determinant 1.*

It is straight-forward to check that axioms of group theory are satisfied by $O(n)$ and $SO(n)$ (see any introductory text on Lie groups, e.g., (Hall, 2015)). In fact they form Lie sub-groups of the group of $n \times n$ invertible matrices $GL(n)$, both of which are manifolds embedded naturally in $\mathbb{R}^{n^2}$.

**Definition 2.** *A Givens rotation, denoted as $R_{i,j}(\theta)$, defines a rotational linear transformation around axes $i$ and $j$, by an angle of $\theta \in [0, 2\pi)$. Formally,*

$$R_{i,j}(\theta) := \begin{pmatrix} \mathcal{I}_{i-1} & & & & \\ & \cos\theta & & -\sin\theta & \\ & & \mathcal{I}_{j-i} & & \\ & \sin\theta & & \cos\theta & \\ & & & & \mathcal{I}_{n-j} \end{pmatrix} \begin{matrix} \\ i \\ \\ j \\ \\ \end{matrix}$$

*where $\mathcal{I}_k$ denotes the $k \times k$ identity matrix. In other words, the Givens rotation matrix $R_{ij}(\theta)$ can be regarded as an identity matrix with four elements at $(i, i)$, $(i, j)$, $(j, i)$, $(j, j)$ replaced.*

Hurwitz (1963) showed the following important result as early as in 1963. The essential idea is similar to Gram-Schmidt process, but specialized to rotational matrices.

**Lemma 1.** *Every element in $SO(n)$ can be written as a product $\prod_{1 \le i < j \le n} R_{i,j}(\theta_{i,j})$, though the decomposition is not unique.*

In our case, to iteratively learn the rotation, we want to take $\theta$ to be small, such as uniformly in $(-\epsilon, \epsilon)$, at each step. While most existing work considers a single Givens rotation at each step, to our best knowledge no published work has considered multiple rotations in one steps, namely with a step rotation composed of $n/2$ **independent** rotations:

$$R = R_{\sigma(1),\sigma(2)}(\theta_{1,2}) R_{\sigma(3),\sigma(4)}(\theta_{3,4}) \dots R_{\sigma(n-1),\sigma(n)}(\theta_{n-1,n}),$$

where $\sigma$ is a **randomly chosen permutation**, $\sigma(i)$ is the value at position $i$, and $\theta_{i,j}$'s are again independent and uniform in $(-\epsilon, \epsilon)$.

The fact that these rotation axes are mutually disjoint ensures that their product can be computed in a parallel fashion. Also the convergence proof of the coordinate descent algorithm relies on estimating the amount of descent in the tangent space, which in turns requires that the rotations commute with one another. This is naturally satisfied by the choice of rotations above.

**Lemma 2.** *The $n/2$ families of Givens rotations $R_{\sigma(1),\sigma(2)}(\theta_{1,2}), \dots, R_{\sigma(n-1),\sigma(n)}(\theta_{n-1,n})$ generate a maximally commuting subgroup within $SO(n)$.*

This follows from the observation that a maximal torus of $SO(n)$ is precisely given by simultaneous rotations along $n/2$ pairs of mutually orthogonal $2d$-planes (see (Bröcker & Tom Dieck, 2013) Chapter 5). For odd $n = 2m + 1$, simply replace $n/2$ by $(n-1)/2$.

Note that both groups $SO(n)$ and $GL(n)$ are non-commutative for all $n > 2$, just as matrix multiplication is in general non-commutative. Unlike $GL(2)$, however, $SO(2)$ is commutative, since it consists of the 1-parameter family of planar rotations.

## 2.3 Block Coordinate Descent Algorithm

The overall objective function, as shown in Eq. (1), needs to be optimized with respect to $R$ and other parameters in $\mathcal{T}$, formally,

$$\underset{R \in SO(n), \mathcal{T}}{\text{minimize}} \quad \mathcal{L}(X) = \mathcal{L}_{ret}(\mathcal{T}(X)) + (1/m)||XR - \phi(XR)||^2, \tag{2}$$

where we omit other parameters in $\mathcal{T}$ since they are not the focus of this paper, though clearly optimizing those parameters needs special techniques such as gradient straight-through estimator (Bengio et al., 2013). Note that, in earlier PQ work, $R$ is either fixed to be $\mathcal{I}_n$, the identity matrix (Chen et al., 2020), or separately optimized in a non-differentiable manner, e.g., using singular value decomposition (Ge et al., 2013), which is equivalent to optimizing only the second term in Eq. (2) with respect to only $R$ and fixing all other parameters.

These earlier approaches do not apply to our case, since we would like to optimize the rotation matrix iteratively and jointly with other parameters. The naive update rule $R \leftarrow R - \alpha \nabla_R \mathcal{L}$, however, does not preserve the requirement that $R \in SO(n)$. A straight-forward remedy is to project the update back to $SO(n)$, such as with SVD or the tangent space exponential map, both of which however require essentially a diagonalization procedure, which is prohibitively expensive. Instead we take a direct approach to ensure that $R$ stays on $SO(n)$ after every iteration.

Consider the Hurwitz parameterization in Lemma 1, since $\theta_{ij} \in \mathbb{R}$, we can treat it as Euclidean optimization, thus a step of gradient descent looks like $R \leftarrow \prod_{1 \leq i < j \leq n} R_{ij}(\theta_{ij} + \alpha \nabla_{\theta_{ij}} \mathcal{L})$. However, full Hurwitz product requires $\Omega(n^2)$ sparse matrix multiplications, which is too expensive. Furthermore the various $\theta_{ij}$ are non-trivially correlated. To address both, we consider **selecting a subset of Givens rotation** $R_{ij}$ at each step, according to the directional derivative as follows.

**Proposition 1.** *Given an objective function $\mathcal{L}$, the (un-normalized) directional derivative $\frac{d}{d\theta}|_{\theta=0} \mathcal{L}(X R_{ij}(\theta))$, $1 \leq i < j \leq n$, is given by*

$$\langle R'_{ij}(0), X^\top \nabla \mathcal{L}(X) \rangle = \text{Trace}(X^\top \nabla \mathcal{L}(X) R'_{ij}(0)^\top) = (\nabla \mathcal{L}(X)^\top X - X^\top \nabla \mathcal{L}(X))_{ij}.$$

*where $(.)_{ij}$ denotes the element at $i$-th row and $j$-th column, and $\langle \cdot, \cdot \rangle$ is the Frobenius inner product.*

*Proof.* Consider the map $h : \mathbb{R} \to \mathbb{R}^{n \times n}$, $h(\theta) = X R_{ij}(\theta)$. By linearity, $h'(0) = X R'_{ij}(0)$. Thus, the chain rule gives

$$\frac{d}{d\theta}\bigg|_{\theta=0} \mathcal{L}(X R_{ij}(\theta)) = \langle \nabla \mathcal{L}(X), X R'_{ij}(0) \rangle.$$

We would like to group $\nabla \mathcal{L}(X)$ and $X$ together. Using the fact that $\langle u, v \rangle = u^\top v$, the right hand side can be written as

$$\nabla \mathcal{L}(X)^\top X R'_{ij}(0) = (X^\top \nabla \mathcal{L}(X))^\top R'_{ij}(0) = \langle X^\top \nabla \mathcal{L}(X), R'_{ij}(0) \rangle. \tag{3}$$

From Definition 2 of Givens rotation $R_{ij}(\theta)$, we see that

$$R'_{ij}(0) = \begin{pmatrix} \mathbf{0}_{i-1} & & & & \\ & \cos'(0) & & -\sin'(0) & \\ & & \mathbf{0}_{j-i} & & \\ & \sin'(0) & & \cos'(0) & \\ & & & & \mathbf{0}_{n-j} \end{pmatrix} \begin{matrix} i \\ \\ j \end{matrix} = \begin{pmatrix} \mathbf{0}_{i-1} & & & & \\ & 0 & & -1 & \\ & & \mathbf{0}_{j-i} & & \\ & 1 & & 0 & \\ & & & & \mathbf{0}_{n-j} \end{pmatrix} \begin{matrix} i \\ \\ j \end{matrix}.$$

---

**Algorithm 1** Bipartite matching in greedy Givens coordinate descent (GCD-G).

---

1: **Input**: $\binom{n}{2}$ directional derivatives $g_{ij} := \frac{d}{d\theta}\mathcal{L}(RR_{ij}(\theta)), 1 \le i < j \le n$.
2: **Output**: $n/2$ disjoint coordinate pairs $\{(i_\ell, j_\ell) : 1 \le \ell \le n/2\}$.
3: Let $S := \{1, \dots, n\}$ be the set of all coordinate axes.
4: **for** $1 \le \ell \le n/2$ **do**
5:     Among indices in $S$, pick $(i_\ell, j_\ell), i_\ell < j_\ell$, that maximizes $\|g_{ij}\|^2$.
6:     Remove $i_\ell$ and $j_\ell$ from $S$: $S \to S \setminus \{i_\ell, j_\ell\}$.
7: **end for**

---

It follows that $\langle R'_{ij}(0), Y \rangle = (Y^\top - Y)_{ij}$ for any square matrix $Y$, which completes the proof by plugging in Eq. (3). Finally to normalize the directional derivative, it needs to be divided by $\|R'_{ij}(0)\| = \sqrt{2}$. □

Given the above directional derivatives for each $R_{ij}(\theta)$, we have a few options to select a subset of $n/2$ **disjoint** $(i, j)$ pairs for $R_{ij}$, listed below in order of increasing computational complexities, where GCD stands for Givens coordinate descent.

- **GCD-R randomly** chooses a bipartite matching and descents along the $n/2$ Givens coordinates, similar to stochastic block coordinate gradient descent (Beck & Tetruashvili, 2013). This can be done efficiently by first shuffling the set of coordinates $\{1, \dots, n\}$.

- **GCD-G** sorts the $ij$ pairs according to the absolute value of its directional derivatives in a decreasing order, and **greedily** picks pairs one by one if they forms a bipartite matching. Since this is our main proposed algorithm, we list the details in Algorithm 1.

- **GCD-S** finds a bipartite matching with maximum sum of edge weights, giving the **steepest** descent. The fastest exact algorithm (Kolmogorov, 2009) for maximal matching in a general weighted graph, has a $O(n^3)$ running time, which is impractical for first-order optimization.

With any of the above options to pick $n/2$ disjoint $ij$ pairs, we can present the full algorithm in Algorithm 2. Moreover, it is worth mentioning that picking non-disjoint $ij$ pairs neither guarantees theoretical convergence as shown in Section 2.5, nor works well in practice (see Section 3.1).

## 2.4 COMPLEXITY ANALYSIS

One iteration of Givens coordinate descent, as shown in Algorithm 2, consists of three major steps: a) computing directional derivatives $A$ in $O(n^3)$ but parallelizable to $O(n)$; b) selecting $n/2$ disjoint pairs in $O(n)$ by random, $O(n^2 \log n)$ by greedy, or $O(n^3)$ by the exact non-parallelizable blossom algorithm (Kolmogorov, 2009); c) applying rotation in $O(2n^2)$ FLOPs by a matrix multiplication between $R$ and an incremental rotation, which is a sparse matrix with $2n$ nonzero elements. Thus, the sequential computational complexities of one iteration for the three proposed algorithms are $O(n^3 + n + 2n^2) = O(n^3)$ for GCD-R, $O(n^3 + n^2 \log n + n^2 + 2n^2) = O(n^3)$ for GCD-G, and $O(n^3 + n^3 + 2n^2) = O(n^3)$ for GCD-S.

Though the time complexity for SVD and Cayley transform's gradient computation is also $O(n^3)$, the derivative computation in the proposed GCD algorithms can be fully parallelized under modern GPUs. Thus GCD-R and GCD-G have parallel runtimes of $O(n^2)$ and $O(n^2 \log n)$ respectively. In contrast, the SVD computation can not be easily parallelized for general dense matrices (Berry et al., 2005), due to its internal QR factorization step (Golub & Van Loan, 1996). So is the Cayley transform gradient computation, since matrix inversion by solving a linear system can not be parallelized.

## 2.5 CONVERGENCE ANALYSIS

In this section we consider gradient descent of a loss function $\mathcal{L} : SO(n) \to \mathbb{R}$. Note that $SO(n)$ is not a convex subset of its natural embedding space $\mathbb{R}^{n^2}$ because convex combinations of two distinct orthogonal matrices are never orthogonal.

---

**Algorithm 2** One iteration of Givens coordinate descent algorithm for rotation matrix learning.

---

1: **Input**: current $n \times n$ rotation matrix $R$, loss function $\mathcal{L} : \mathbb{R}^{m \times n} \to \mathbb{R}$, learning rate $\lambda$.
2: Get a batch of input $X \in \mathbb{R}^{m \times n}$ and run forward pass using $X' = XR$.
3: Run backward pass to get the gradient $G = \nabla_R \mathcal{L}(XR)$ and compute $A := G^\top R - R^\top G$.
4: **for** $1 \leq i < j \leq n$ **do**
5:     Gather $A_{ij}$ as the directional derivative $g_{ij} := \frac{1}{\sqrt{2}} \frac{d}{d\theta}|_{\theta=0} \mathcal{L}(RR_{ij}(\theta))$ (see Proposition 1).
6: **end for**
7: Select $n/2$ disjoint pairs $\{(i_\ell, j_\ell) : l\}$ by random, greedy (see Algo. 1) or steepest options.
8: $R \leftarrow R \prod_{\ell=1}^{n/2} R_{i_\ell, j_\ell}(-\lambda g_{i_\ell, j_\ell})$.

---

As usual, convergence results require some assumptions about the domain and the function. Before stating them, first we introduce some definitions, specialized to the space $SO(n)$ of orthogonal matrices with determinant 1.

For all below lemmas, theorem and corollary, see Appendix for a complete proof.

**Definition 3.** *The **tangent space** of $SO(n) \subset \mathbb{R}^{n^2}$ at the identity $\mathcal{I}_n$ is given by the space of anti-symmetric matrices $\mathfrak{so}(n) := \{g \in \mathbb{R}^{n \times n} : g + g^\top = 0\}$. That of $GL(n)$ is simply $\mathfrak{gl}(n) := \mathbb{R}^{n^2}$.*

**Definition 4.** *A **geodesic** on $SO(n)$ is a curve of the form $\xi(t) = G^* \exp(tg)$ for a starting point $G^* \in SO(n)$ and a direction given by some $g \in \mathfrak{so}(n)$; $\exp$ is the matrix exponential.*

**Definition 5.** *A function $\mathcal{L} : SO(n) \to \mathbb{R}$ is called **geodesically convex** if for any geodesic $\xi$, $\mathcal{L} \circ \xi : [0, 1] \to \mathbb{R}$ is a convex function.*

We make two assumptions on $\mathcal{L}$ in order to prove global convergence results for the random Givens coordinate descent method (GCD-R) on $SO(n)$. The convergence of GCD-G and GCD-S follows by simple comparison with GCD-R at each step.

**Assumption 1.** *The objective function $\mathcal{L}$ is geodesically convex on the level set of the manifold $S(G_0) := \{G \in SO(n) : \mathcal{L}(G) < \mathcal{L}(G_0)\}$ and has a unique global minimum at $G^*$.*

**Assumption 2.** *(Global Lipschitz) For some $\eta > 0$ and all $G \in S(G_0)$, $g \in \mathfrak{so}(n)$:*

$$\mathcal{L}(G \exp(g)) \leq \mathcal{L}(G) + \langle \nabla \mathcal{L}(G), g \rangle + \frac{\eta}{2} \|g\|^2. \tag{4}$$

Assumption 2 leads to the following Lemma

**Lemma 3.** *For learning rate $\alpha = \frac{1}{\eta}$, the descent is sufficiently large for each step:*

$$\mathbb{E}[\mathcal{L}(G_k) - \mathcal{L}(G_{k+1})|G_k] \geq \frac{1}{(n-1)\eta} \|\nabla \mathcal{L}(G_k)\|^2.$$

Lemma 3.4-3.6 of (Beck & Tetruashvili, 2013) combined with the above lemma and Assumption 1 finally yield the following Theorem:

**Theorem 1.** *Let $\{G_k\}_{k \geq 0}$ be the sequence generated by the GCD-R method, and let $D_n := n\pi$ be the diameter of $SO(n)$. With the two assumptions above,*

$$\mathcal{L}(G_k) - \mathcal{L}(G^*) \leq \frac{1}{k(D_n^2(n-1)\eta)^{-1} + (\mathcal{L}(G_0) - \mathcal{L}(G^*))^{-1}}.$$

*In particular, the value of $\mathcal{L}(G_k)$ approaches that of the minimum, $\mathcal{L}(G^*)$, at a sub-linear rate.*

By verifying Assumptions 1 and 2 for individual functions $F$, we have

**Corollary 1.** *Let the **joint** rotational product quantization learning objective be of the form*

$$\tilde{F}(R, \mathcal{C}) = F\left(R^\top \bigoplus_{i=1}^{D} u_i\right) \quad s.t. \quad u_i \in \mathcal{C}_i \subset \mathbb{R}^{n/D} \quad and \quad \|\mathcal{C}_i\| = K,$$

*where $\bigoplus_{i=1}^{D} u_i := (u_1, \dots, u_D)$ stands for vector concatenation. Consider the specialization to a convex $F$, such as, $F(x) = \langle w, x \rangle$ (dot product) or $F(x) = \frac{\langle w, x \rangle}{\|w\|\|x\|}$ (cosine similarity). If we fix the value of $w \in \mathbb{R}^n$ as well as the choice of quantization components $u_i$, gradient descent applied to the map $\mathcal{L} : R \mapsto \tilde{F}(R, \mathcal{C})$ converges to the global minimum, provided the latter is unique.*

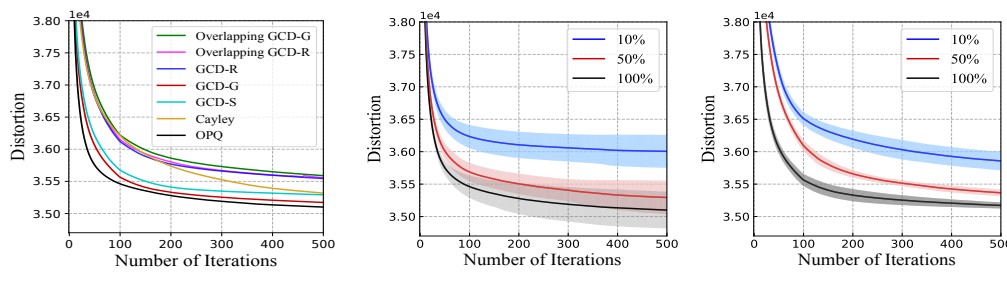

(a) Convergence comparative results.    (b) OPQ on various datasets.    (c) GCD-G on various datasets.

Figure 2: Performance of the proposed GCD algorithms in the SIFT1M dataset (Lowe, 2004).

## 3 EXPERIMENTS

We evaluate the convergence performance on fixed embeddings in Section 3.1, and the overall embedding retrieval performance on end-to-end trainable embedding indexes in Section 3.2. We implement the proposed GCD algorithm in Tensorflow 1.15, and perform all experiments in a single machine with 4 Nvidia 16G V100 GPU cards.

### 3.1 FIXED EMBEDDINGS

**Dataset and Setup.** To make sure the proposed GCD algorithms practically converge, we first evaluate, for a given set of embeddings from SIFT1M (Lowe, 2004), whether the GCD algorithms can converge similarly as the original OPQ algorithm (Ge et al., 2013), which alternates between k-means for PQ centroid learning and SVD projection for optimal rotation. Our GCD algorithms simply replace the latter SVD step by a number (5 in this experiment) of Givens coordinate descent iterations with learning rate $\lambda$=0.0001 as shown in Algorithm 2. Similarly, Cayley method (Helfrich et al., 2018; Casado & Martínez-Rubio, 2019) also replaces the SVD steps by a number of Cayley parameter updates. All methods are evaluated by quantization distortion, which is the square of L2 distance between the original embedding and quantized embedding, averaged over all examples.

Figure 2a shows the comparative results between OPQ, Cayley method and the proposed GCD algorithms. Specifically, the proposed GCD-G and GCD-S can converge as well as the original OPQ algorithm, as the convergence curves are similar. The previously widely used method, Cayley transform, does not converge as fast as the proposed GCD algorithms, potentially due to the varying scales of the transformed gradient. It also shows the results of two other alternatives: overlapping GCD-G and overlapping GCD-R, which do not enforce the $(i, j)$ pairs to be disjoint as we discussed in Section 2.2. The comparison shows the necessity of enforcing the disjoint selection, since otherwise GCD-G does not converge well, though it does not affect GCD-R much. But GCD-R in general does not work as well as GCD-G, which indicates the necessity of picking the steeper descent directions. Moreover, we can observe that GCD-G converges similarly as GCD-S, which indicates that a greedy bipartite matching algorithm as given in Algorithm 1 may be good enough in practice.

Figures 2b and 2c further compare the convergence performance between GCD-G and OPQ, by averaging over 10 runs and varying the size of data. We can make two important observations: a) GCD-G converges much more stably than OPQ, as the curves show significantly lower variances; b) GCD-G converges better for smaller size of data, e.g., for 10% of the data. It indicates that GCD-G potentially works better in the stochastic descent scenario where the batch size is usually small.

### 3.2 END-TO-END TRAINABLE EMBEDDING INDEXES

**Dataset and Setup.** In this section, we evaluate GCD-S and GCD-R performances in an industrial search click log data where a user input query is used to retrieve items, and two public datasets of MovieLens (Harper & Konstan, 2015) and Amazon Books (He & McAuley, 2016) where user historical behaviors are used to retrieve next behavior. The industrial dataset contains 9,989,135 training examples with 1,031,583 unique queries and 1,541,673 unique items, which is subsampled about 2% from our full dataset, but enough for training a reasonably well search retrieval model. We evaluate the effectiveness of the learned rotation matrix by quantization distortion, precision@$k$

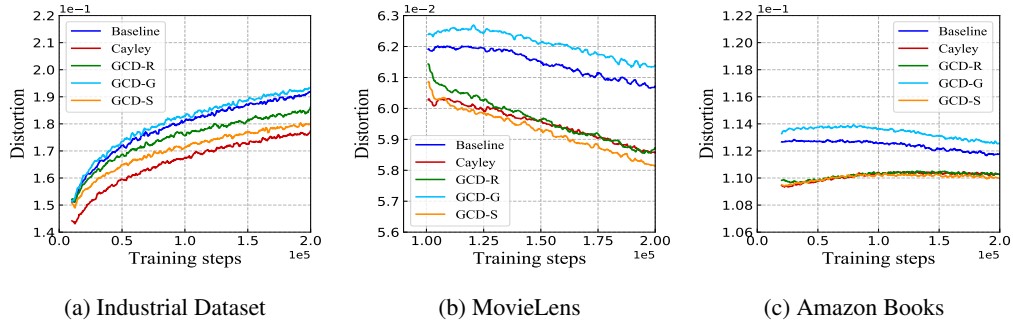

|  | (a) Industrial Dataset | (b) MovieLens | (c) Amazon Books |

Figure 3: Comparison of quantization distortion reduction on end-to-end trainable embedding indexes.

Table 1: Comparison of retrieval quality metrics on end-to-end trainable embedding indexes.

|  | Industrial Dataset | | MovieLens | | Amazon Books | |
| --- | --- | --- | --- | --- | --- | --- |
|  | p@100 | r@100 | p@100 | r@100 | p@100 | r@100 |
| Baseline | 2.43% | 51.29% | 7.78% | 35.53% | 0.74% | 5.91% |
| Cayley | 2.44% | 51.37% | 7.73% | 35.59% | 0.74% | 5.88% |
| GCD-R | 2.43% | 51.28% | 7.82% | 35.78% | 0.73% | 5.88% |
| GCD-G | 2.43% | 51.37% | 7.83% | 35.89% | 0.73% | 5.81% |
| GCD-S | **2.44%** | **51.43%** | **7.94%** | **36.35%** | 0.74% | 5.91% |
| GCD-S p-value w/ baseline | 0.054 | 0.012 | 0.145 | 0.024 | 0.702 | 0.925 |

(p@$k$) and recall@$k$ (r@$k$) metrics, which are standard retrieval quality metrics. In our experiments, we choose $k$=100. Specifically, for each query, we retrieve from embedding indexes a set of top $k$ items, which is then compared with the ground truth set of items to calculate the precision and recall. We perform ANOVA test to evaluate the p-value for all comparisons. We implement a two-tower retrieval model (Zhang et al., 2021) with cosine scoring, hinge loss of margin 0.1 and embedding size 512. At the beginning, a number of 10,000 warmup steps are performed before applying the indexing layer as shown in Figure 1. Then a number of 8,192 examples are collected to warm start the PQ centroids and rotation matrix with 200 iterations of OPQ algorithm. The baseline method uses this initial rotation matrix and stops updating it, while the other methods continue updating it.

As shown in Figure 3, in all three datasets, the proposed GCD algorithms and Cayley method clearly outperform the baseline in terms of quantization distortion, which consequentially translates into the improvement in retrieval quality metrics, measured by precision@100 and recall@100 as shown in Table 1. However, the Amazon Book dataset is an exception. We suspect the dataset may have subtle distribution, thus the small quantization distortion reduction does not result in improvements on retrieval quality metrics. Finally, we observe that the three proposed GCD algorithms consistently with theoretical analysis – GCD-R and GCD-S can be regarded as the lower bound and upper bound of GCD methods, with GCD-G in between. The GCD-S with steepest Givens rotations picked at each step, generally shows the best performance in both quantization distortion and retrieval metrics, especially better than the widely known Cayley method in literature.

## 4 CONCLUSION

In this paper, we have proposed a novel method called Givens coordinate descent (GCD) to iteratively learn a rotation matrix, in the context of end-to-end trainable PQ based embedding indexes. Based on a well studied mathematical result that any orthogonal matrix can be written as a product of Givens rotations, the proposed GCD algorithm picks $n/2$ maximally commuting Givens rotations to update the learning rotation matrix by their directional derivatives. We also provide complexity analysis and prove convergence of the proposed algorithm under convexity and other mild assumptions. Experimental results show that the proposed method converges more stably than previous SVD based and Cayley transform method, and it significantly improves retrieval metrics in the end-to-end training of embedding indexes. Our algorithm is also applicable to other scenarios in any neural network training where learning a rotation matrix is potentially helpful.

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

## A CONVERGENCE RESULTS

In this section we prove the results stated in Section 2.5.

**Lemma 1.** *For learning rate $\alpha = \frac{1}{L}$, the descent is sufficiently large for each step:*

$$\mathbb{E}[F(x_k) - F(x_{k+1})|x_k] \geq \frac{1}{2(n-1)L}\|\nabla F(x_k)\|^2.$$

*Proof.* Recall that each step of RGCGD is given by $x_{k+1} = x_k \cdot G$ where $G = \prod_{i=1}^{n/2} g_{\sigma(2i-1),\sigma(2i)}(\theta_i)$, where $\sigma \in S_n$ is a uniformly random permutation. Furthermore since the $g_{\sigma(2i-1),\sigma(2i)}$'s in the product are along disjoint axes, they commute with one another:

$$g_{\sigma(2i-1),\sigma(2i)}(\theta_i)g_{\sigma(2j-1),\sigma(2j)}(\theta_j) = g_{\sigma(2j-1),\sigma(2j)}(\theta_j)g_{\sigma(2i-1),\sigma(2i)}(\theta_i), \tag{5}$$

we can write them as $g_{i,j}(\theta) = \exp(\frac{\theta}{\sqrt{2}}(e_i \wedge e_j))$, where the $\sqrt{2}$ factor accounts for the fact that $\|e_i \wedge e_j\|^2 = 2$ under Euclidean norm.

Thus let's define $s \in \mathfrak{so}(n) := T_{\mathcal{I}}SO(n)$ by

$$s := \sum_{i=1}^{n/2} \frac{\theta_i}{\sqrt{n}} e_{\sigma(2i-1)} \wedge e_{\sigma(2i)},$$

$$x_{k+1} = R_{x_k}(\alpha s) = x_k \cdot \exp(\alpha s) = x_k \cdot \exp(\alpha \sum_{i=1}^{n/2} \frac{\theta_i}{\sqrt{2}} e_{\sigma(2i-1)} \wedge e_{\sigma(2i)}).$$

In fact, the $\theta_i$'s are projections of the full gradient $\nabla \tilde{F}(x)$, for $\tilde{F}(x) := F(x_k \cdot x)$,

$$\theta_{i,j} := -\langle \nabla \tilde{F}(\mathcal{I}), \frac{1}{\sqrt{2}} e_i \wedge e_j \rangle, \text{ for } 1 \leq i < j \leq n.$$

So from (4), and orthogonality of $\frac{1}{\sqrt{2}} e_{\sigma(2i-1)} \wedge e_{\sigma(2i)}$, we have

$$F(x_{k+1}) \leq F(x_k) - \alpha\|s\|^2 + \frac{L\alpha^2}{2}\|s\|^2. \tag{6}$$

Now choose $\alpha = \frac{1}{L}$. We easily get

$$F(x_{k+1}) - F(x_k) \leq -\frac{1}{2L}\|s\|^2. \tag{7}$$

Finally recall that $s \in \mathfrak{so}(n)$ is a random vector with randomness given by the uniform $\sigma \in S_n$. By orthogonality and the definition of $\theta_i$ above, we have

$$\mathbb{E}\|s\|^2 = \frac{1}{n!}\sum_{\sigma \in S_n}\sum_{i=1}^{n/2}\theta_{\sigma(2i-1),\sigma(2i)}^2 = \frac{1}{n-1}\sum_{1\leq i<j\leq n}\theta_{i,j}^2 = \frac{\|\nabla\tilde{F}(0)\|^2}{n-1} = \frac{\|\nabla F(x_k)\|^2}{n-1}. \tag{8}$$

Taking expectation of both sides of (7):

$$\mathbb{E}[F(x_{k+1})|x_k] \leq F(x_k) - \frac{1}{2(n-1)L}\|\nabla F(x_k)\|^2. \tag{9}$$

$\square$

Next we recall Lemma 3.5 from Beck & Tetruashvili (2013)

**Lemma 2.** *Let $\{A_k\}_{k\geq 0}$ be a non-negative sequence of real numbers satisfying $A_k - A_{k+1} \geq \eta A_k^2$, $k = 0, 1, \ldots$, and $A_0 \leq \frac{1}{m\eta}$ for positive $\eta$ and $m$. Then*

$$A_k \leq \frac{1}{\eta(k+m)}, \text{ for } k = 0, 1, \ldots \tag{10}$$

$$A_k \leq \frac{1}{\eta k}, \text{ for } k = 1, 2, \ldots \tag{11}$$

This along with Assumption 1 leads to global convergence result below (combining Lemma 3.4 and Theorem 3.6 of Beck & Tetruashvili (2013)):

**Theorem 1.** *Let $\{x_k\}_{k \geq 0}$ be the sequence generated by the RGCGD method. With the two assumptions above,*

$$F(x_k) - F(x^*) \leq \frac{1}{2kR^2(x_0)(n-1)L + (F(x_0) - F(x^*))^{-1}}. \tag{12}$$

*In particular, the value of $F(x_k)$ approaches that of the minimum, $F(x^*)$, at a sub-linear rate.*

*Proof.* We can write $x_k = R_{x^*}(s)$, where $s = \gamma'(0)$ and $\gamma : [0,1] \to S(x_0) \subset SO(n)$ is a constant speed geodesic path from $x^*$ to $x_k$. This means that $\|\gamma'(t)\|$ is constant for all $t$.

By geodesic convexity, $F \circ \gamma : [0,1] \to \mathbb{R}$ is convex, therefore

$$F(x_k) - F(x^*) \leq (F \circ \gamma)'(0)\langle \nabla F(x_k), \gamma'(0)\rangle.$$

Since $\gamma'(0) = d(x_k, x^*)$, the geodesic distance between $x_k$ and $x^*$, by the constant speed parametrization of $\gamma$, Cauchy-Schwarz inequality then implies

$$F(x_k) - F(x^*) \leq \|\nabla F(x_k)\| d(x_k, x^*) \leq R(x_0)\|\nabla F(x_k)\|.$$

This combined with Lemma 1 gives

$$\mathbb{E}[F(x_k) - F(x_{k+1}|x_k] \geq \frac{1}{2R^2(x_0)(n-1)L}(F(x_k) - F(x^*))^2.$$

Now consider the Taylor expansion around the global minimum $x^*$. Since the gradient vanishes, the Lipschitz condition (4) simplifies to

$$F(x_k) - F(x^*) \leq \frac{L\|s\|^2}{2} \leq \frac{L}{2}R^2(x_0).$$

For the last inequality, we again use $\gamma'(0) = d(x_k, x^*) \leq R(x_0)$ by definition of the latter.

So letting $A_k = E[F(x_k) - F(x^*)]$, $\eta = \frac{1}{2R^2(x_0)(n-1)L}$, and $m = \frac{1}{\eta(F(x_0) - F(x^*))}$, Lemma 2 implies that

$$F(x_k) - F(x^*) \leq \frac{1}{(k+m)\eta} = \frac{1}{2kR^2(x_0)(n-1)L + (F(x_0) - F(x^*))^{-1}}.$$

$\square$

**Corollary 1.** *Let the **joint** rotational product quantization learning objective be given by*

$$L(w, R, \mathcal{C}) = F(w)(R \bigoplus_{i=1}^{D} u_i) \quad s.t. \quad u_i \in \mathcal{C}_i \subset \mathbb{R}^{n/D} \quad and \quad \|\mathcal{C}_i\| = K. \tag{13}$$

*Consider the specialization of $w \in \mathbb{R}^n$ and $F(w)(x) = \langle w, x\rangle$ (dot product) or $F(w)(x) = \frac{\langle w, x\rangle}{\|w\|\|x\|}$ (cosine similarity). If we fix the value of $w \in W$ as well as the choice of quantization components $u_i$, gradient descent applied to the map $\tilde{F} : R \mapsto L(w, R, C)$ converges to the global minimum, provided the latter is unique.*

*Proof.* In the dot product case, $\tilde{F}(R) = \langle w, Rx\rangle$. If there is a unique global minimizer, it can be given by $UV^\perp$ from the following singular value decomposition:

$$wx^\perp = USV^\perp.$$

First we verify Assumption 1, namely that the following function is convex:

$$t \mapsto \tilde{F}(G_0 \exp(tA)) = \langle w, G_0 \exp(tA)x\rangle.$$

Indeed this follows immediately from diagonalizing $A$, which yields a linear combination of convex functions of $t$.

For assumption 2, recall we need to show $\tilde{F}(G_0 \exp(A)) \leq \tilde{F}(G_0) + \langle \nabla \tilde{F}(G_0), A \rangle + \frac{L}{2}\|A\|^2$, for some $L > 0$. By differentiating $\langle w, G_0 \exp(tA)x \rangle$ at $t = 0$, we see that

$$\frac{d}{dt}|_{t=0}\tilde{F}(G_0 \exp(tA)) = \langle \nabla \tilde{F}(G_0), A \rangle$$

$$\tilde{F}(G_0 \exp(A)) - \tilde{F}(G_0) = \langle w, G_0(\exp(A) - \mathcal{I})x \rangle = \langle w, G_0(A + \frac{1}{2}A^2 + \ldots)x \rangle$$

$$= \langle \nabla \tilde{F}(G_0), A \rangle + \frac{1}{2}\langle \nabla \tilde{F}(G_0), A^2 \rangle \sum_{k=2}^{\infty} \frac{1}{k!}\langle w, A^k x \rangle.$$

For $\|A\|$ small, the last term can be bounded by $e^{\|A\|} - 1 - \|A\|$, which is $O(\|A\|^2)$. For larger $\|A\|$, say $> C$, we can choose $L$ large enough so that $\frac{LC^2}{2} > \sup F$.

The cosine similarity case follows immediately from the dot product case since the denominator is a constant for fixed $w$ and $\bigoplus_{i=1}^{D} u_i$, thanks to the fact that $\|Rx\| = \|x\|$. $\qquad\square$

**Corollary 2.** *Under the two assumptions 1 and 2, the steepest Givens coordinate gradient descent (SGCGD) method also converges to its unique global minimum.*

*Proof.* By the choice of coordinates, each step of SGCGD is dominated by the corresponding step of RGCGD based at the same point $x \in SO(n)$, in other words, the estimate (12) continues to hold. $\quad\square$

## B  EMPIRICAL RUNNING TIME

Figure 4 shows the comparison of per step runtime cost between GCD methods and Cayley transform for standalone QP training, with batch size chosen to be 1. In Figure 4a, for a fair comparison, we only compare the GCD-R method with Cayley transform, since we have not implemented GCD-G and GCD-S fully in GPU operators yet. Note that the GCD-G method needs a sorting operator implemented in GPU (see step 5 in Algorithm 1), which is possible but needs a customized GPU operator implemented in addition to the original TensorFlow release. On the other hand for Cayley transform, while the gradient propagation and matrix multiplication are all computed on GPU, the matrix inversion operation unfortunately cannot be parallelized, thus effectively runs on CPU. As expected from the complexity analysis in Section 2.4, GCD-R slows down quadratically in the embedding dimension $n$, whereas Cayley incurs an asymptotic $O(n^3)$ cost per step. To further explore the empirical computation cost in a completely fair setup, we compare the methods all in CPU. As shown in Figure 4b, GCD-G and GCD-R run much faster than the Cayley method, with about 5 times and 50 times faster running speed, respectively, which roughly reflect the total number of floating point computations in each method.

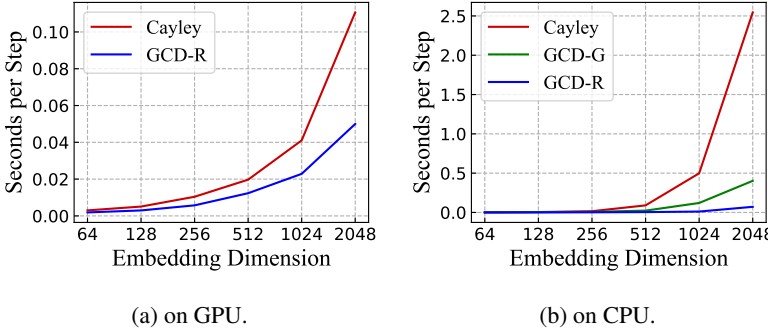

(a) on GPU.    (b) on CPU.

Figure 4: Empirical runtime comparison between GCD methods and Cayley transform method.

