# OpenReview forum: "Givens Coordinate Descent Methods for Rotation Matrix Learning in Trainable Embedding Indexes"
_ICLR.cc/2022/Conference — ICLR 2022 Poster_

### Official Review · Reviewer_Qvs3 · 2021-10-21

**Correctness:** 3
**Technical Novelty And Significance:** 3
**Empirical Novelty And Significance:** 2
**Recommendation:** 6
**Confidence:** 4

**Main Review:**

There is a mistake in Lemma 1 and 2. There must be n(n-1)/2 components instead of n/2 as claimed in the Lemmas. Take SO(3) for example, any minimal representation of SO(3) needs 3 numbers. But n/2=1.5. The value $n/2$ appears wrongly in a few places thereafter.

While I think there is no major issue with mathematics in the paper apart from the mistake in Lemma 1 and 2, I think the authors may have overlooked the power of the Lie algebra so(n) of rotation matrices that can give them a few benefits. One can view their representation (or let us call it the Givens representation) of a small rotation matrix $R$ as $R = \prod_{i<j} expm( \theta_{i,j} B_{i,j} )$ (ignoring the sigma terms for ease of discussion) where each $B_{i,j}$ is a matrix of zeros everywhere except that $B_{i,j} = 1$ and $B_{j,i} = -1$, and $expm(X) = \sum_{k>=0} (1/k!) X^k$ is the matrix exponential function. Here, $B_{i,j}$ is equivalent to a bivector such that the set of the $n(n-1)/2$ bivectors forms so(n). Their representation is similar to, but not the same as, the common Lie algebra representation of a small rotation matrix $R = expm(\sum_{i<j} \theta_{i,j} B_{i,j})$.

With the Lie algebra representation, given that R is small, the computation of expm() can be approximated by using the definition above but restricting to a first few terms, e.g. k <= 2 or 3. There is no need to do an expensive diagonalisation (I think the authors should elaborate the need of diagonalisation - it is not clear from the paper).

Secondly, in doing so, line 8 of Algorithm 2 can be replaced by a simpler and faster to compute formula $R \leftarrow R expm(\sum_{i<j} \theta_{i,j} B_{i,j})$, again taking the advantage of approximating $expm()$, eliminating the need to restrict to selected coordinates for gradient descent. There is a concern of whether doing this way would make updated rotation matrices falling off SO(n). However, one can argue that many standard functions implemented on modern machines have already incurred some level of approximation. Hence, the question is not about whether the updated matrices fall off SO(n), but about when they fall off. I would argue that if done right, adding a loss term for penalizing when the rotation is off the manifold for example, you may not have to correct the rotation matrix (via SVD to project it down to SO(n)) in every training step.

Thirdly, in the Givens representation I would not be too sure if bounding each element $\theta_{i<j}$ with $-\epsilon < \theta_{i<j} < \epsilon$ is good enough for small $\epsilon$. Each Givens rotation can incur a small error. The product of all the Givens rotations can make the error on R rather large in a non-linear way. With the Lie algebra representation, we know $R$ is a function of a linear sum. Hence, how large R can be compared to $\epsilon$ is much more predictable.

Finally, the main challenge of the Lie algebra representation compared to the Givens representation is, in my view, computing the partial derivatives efficiently. But here we can take advantage of our small rotation assumption as well. According to:

Rossmann, Wulf (2002), Lie Groups – An Introduction Through Linear Groups, Oxford Graduate Texts in Mathematics, Oxford Science Publications, ISBN 0-19-859683-9

in theorem 5, section 1.2, one can mathematically compute:

$(d/dt) expm(X(t)) = expm(X(t)) \sum_{k>=0} (-1)^k/((k+1)!) (ad_X)^k (d(X(t))/dt)$,

where $ad_X(Y) = XY-YX$ is the adjoint operator. Applying this formula to the Lie algebra representation above, restricting to k <= 2 or 3, making use of the fact that $B_{i<j}$ are constant and very sparse, one may end up with a rather efficient approximation of $dR/d_{\theta_{i<j}}$ for all pairs $i<j$.

The experiments are sufficient to me. However, the improvement gains when switching to the proposed approaches are rather small (i.e. distortion after 500 iterations reduced from about 3.55e-4 to about 3.50e-4 (Fig 2a), and less than 1% p-value improvements in Tab. 1) that sparks a question of statistical significance.

**Summary Of The Paper:**

The paper address rotation matrix learning during product quantization in modern ANN embedding search systems. The main contribution is addressing rotation matrix learning via gradient descent of small rotation updates. The approach relies on the decomposition of any small rotation matrix into a product of Given rotations, so that partial derivatives can be obtained in parallel, but the product causes the computation of the rotation matrix itself to be slow, in O(n^2) matrix multiplications, hence the need to select a subset of coordinates and do coordinate descent. Experiments on product quantization show a marginal improvement in results over existing approaches OPQ and Cayley.

**Summary Of The Review:**

I do not know much about product quantization but I know a bit about rotation matrices. For me, necessary experiments to prove the point of the paper in product quantization are presented, but the gain seems marginal.

Regarding rotation matrix learning, I think the authors may have overlooked the standard approach of projecting rotations to a tangent space/Lie algebra. In this particular problem when the rotations are small, Lie algebra can be very helpful. The paper would be a lot more interesting if there were deeper treatment on this front.

---

> ### Author Response · Authors · 2021-11-16
> **Point to Point Responses**
>
> Thank you very much for the very detailed review and constructive feedback.
>
> 1. Regarding n/2 versus n choose 2 (n(n-1)/2) rotations, Lemma 1 stated that every rotation can be expressed as a product of indeed n(n-1)/2 Givens rotations, since the product is over indices 1 \leq i < j \leq n, which consists of all distinct (unordered) pairs. In Lemma 2, we claim that the span of n/2 mutually orthogonal rotations is a maximally commuting subgroup (also known as a maximal torus) with SO(n), rather than the entire SO(n) itself. Here we indeed forgot to mention that n is assumed to be an even number (this has now been added to the new revision, at the beginning of section 2.2. For odd n, n/2 should be replaced with (n-1)/2; thus for instance the maximal tori of SO(3) consists of 1-dimensional rotations.
>
> 2. Regarding your suggestion of using the Lie algebra (tangent space) exponential approximation to replace the Givens coordinate descent method, indeed that was the first idea that came to mind. For instance one could compute partial derivatives with respect to each Lie algebra basis element at every step. However we were indeed afraid of the error that can arise from only a few terms of the Taylor approximation. This line of research has also been explored in Lezcano-Casado & Martinez-Rubio. "Cheap Orthogonal Constraints in Neural Networks: A Simple Parametrization of the Orthogonal and Unitary Group". ICML 2019. The goal there is to approximate the matrix exponential with a rational function, of which Cayley is a special case. While the Pade approximant yields an exact orthogonal matrix, the higher order rational function incurs additional computation, making it less attractive for us in practice.
>
>  The so-called diagonalization is actually not expensive at all. The idea is to choose the best n/2 disjoint coordinate pairs with highest partial derivatives. The greedy algorithm turns out to be O(n^2 log n), since one could do a global sorting of the n choose 2 pairs first.
> We respectfully disagree on the claimcontention that Givens rotation introduces more error than Lie algebra exponentiation. Indeed the product of rotation matrices in floating calculation can preserve the orthogonality property remarkably well, even after thousands of steps. This was observed empirically during our experiment, through the monitoring of Frobenius Norm(M^t M - I), which is in the order of 10^-7 even after thousands of steps.
>
> 3. The point of bounding the Givens angles by \epslion is not out of concern of approximation error. Rather this is just the step size of gradient descent algorithm. We respectfully disagree on the claim that Givens rotation introduces more error than Lie algebra exponentiation. Indeed the product of rotation matrices in floating calculation can preserve the orthogonality property remarkably well, even after thousands of steps. This was observed empirically during our experiment, through the monitoring of Frobenius Norm(M^t M - I), which is in the order of 10^-7 even after thousands of steps.
>
> 4. The derivative of the exponential of Lie algebra element that you provided is indeed a nice formula. To preserve orthogonality after each gradient descent using it, however, one would still need a method to project the parameters back to SO(n). Similar objective can be accomplished more simply through the Cayley transform or the slightly more complicated Pade approximation method, without the need to project back onto SO(n). In fact, even taking the derivative of the Cayley transform A \mapsto (I - A) (I + A)^{-1} itself is much simpler than that of expm(A). Thus we did not consider this more general approach. Our aim is to preserve the orthogonality structure exactly (at least in theory) with relatively low computation overhead,  since our training tends to run for hundreds of thousands of steps.

---

> > ### Comment · Reviewer_Qvs3 · 2021-11-17
> > **Glad that we are converging.**
> >
> > I would like to thank the authors for their response.
> >
> > 1. Indeed, I overlooked the fact that you restricted the rotation space in each training step further to a $n/2$-dimensional subgroup of $SO(n)$. Thanks for pointing out.
> >
> > 2. I am glad that you find the idea of using Lie algebra useful. It came from the fear, while reading the initial version of the paper, that you are wasting too many dimensions at each training step by restricting to just the maximally commuting subgroup. I do not know the typical value of $n$ but I would imagine that, for example, for an embedding space of 50 dimensions. $SO(n)$ is 1225-dimensional but at every training step here although the algorithm in the paper computes 1225 partial derivatives with respect to the product quantisation function, it only deals with 50 dimensions to improve the rotation. It might require a very large number of training steps to converge.
> >
> > 3. Indeed you restrict to commuting subgroups of $SO(n)$ then what I raised earlier was irrelevant. I was under the impression that you addressed $SO(n)$ as a whole rather than just the commutative subgroup. My apology.
> >
> > 4. Not sure I understand the points you raised in here. As I mentioned earlier, sure one might need to project the parameters back to $SO(n)$. However, given that the role of the rotation matrix in this case is to make the rotated space as easily quantised as possible, do we really need precise rotation matrices living in $SO(n)$ or do we just need an invertible matrix that we can quickly encode (rotate) and decode (de-rotate) embeddings? If the answer leans towards the latter then why not "relax" the rotation matrix at bit, accept some error, and place an additional loss (e.g. $||R R^T-I||_F$) so that the task of adjusting the rotation matrix becomes part of gradient descent? The gain is then you are free to optimise in the $n(n-1)/2$-dimensional space of $SO(n)$ rather than restricting to coordinate descent. Regarding Cayley transform and Pade approximants, I did not mention them in my feedback earlier. You pointed out in the paper that the Cayley transform is not GPU-friendly. Pade approximants may require solving a linear system. I can see they are not GPU friendly either, but maybe better than Cayley transform.

---

> > > ### Author Response · Authors · 2021-11-21
> > > **Addressing the remaining issues**
> > >
> > > Thank you for your thoughtful feedback to each of the explanations. Below are our renewed responses.
> > >
> > > 1. Aknowledged.
> > >
> > > 2. For 50d rotation matrix, we indeed compute 2500 partial derivatives, treating it as if it’s an arbitrary point in the full 2500d Euclidean space. Then the individual directional derivatives can be basically “read off” from these partials, with minimal additional computation.
> > >
> > > 3. No problem! In typical block coordinate descent, there is no restriction on which coordinates can be grouped together. Will emphasize it more in the revision.
> > >
> > > 4. Thank you for the explicit suggestion of using an L^2 regularization term to enforce orthogonality approximately. We agree that as an end to end learning task, this simpler approach is likely sufficient. In standalone product quantization, however, the aim is to preserve the distance between queries and quantized item embeddings exactly, after both are fixed (through the fixing of trainable model parameters). The standalone product quantization step is used primarily to compress the storage of item embeddings, as well as allow faster retrieval, see e.g., “Tiezheng Ge, Kaiming He, Qifa Ke, and Jian Sun. Optimized product quantization for approximate nearest neighbor search”. Thus in the standalone setting, we really need an exact rotation matrix to preserve the pairwise distance from the trained model as much as possible.
> > >
> > > Regarding Cayley and Pade approach, we agree that parallelization remains an obstacle due to the need for matrix inversion. They also suffer from ill-conditioning problems, when the inverted matrix has eigenvalues close to 0. That is why we believe our "moving frame" coordinate descent approach is more suitable for global optimization.

---

> > > > ### Comment · Reviewer_Qvs3 · 2021-11-22
> > > > **Concluding**
> > > >
> > > > Well, your point of exact distance preserving should have been emphasized in the paper, it would have made the motivation stronger.
> > > >
> > > > There are ways to bend my suggestion to go along with it. You can do rotation alignment after training is done. Or you can have an increasing coefficient on the L2 regularization term once your main loss function becomes stable.
> > > >
> > > > Happy to update my ratings once the key points in this discussion have been updated to the paper.
> > > >
> > > > Thank for the paper and for the valuable discussion.

---

> > > > > ### Author Response · Authors · 2021-11-23
> > > > > **Revision summary**
> > > > >
> > > > > Thank you for the concluding comments.
> > > > >
> > > > > Rotation alignment is similar to the periodic svd approach used by the OPQ baseline.
> > > > >
> > > > >  We have added a paragraph to discuss the possibility of using L2 regularization to enforce approximate orthogonality (the last paragraph in Section 1.2, right before Section 1).
> > > > >
> > > > > A paragraph before Lemma 2 was added to emphasize the special choice of rotation axes in each step of GCD.

---

> > > > > > ### Comment · Reviewer_Qvs3 · 2021-11-29
> > > > > > **Thank you!**
> > > > > >
> > > > > > I've read the revised version of the paper. Thank you for the updates. While the paper looks better now, the revisions do not significantly alter my rating.
> > > > > >
> > > > > > Thank you for the paper. It was a good read.

---

### Official Review · Reviewer_HiTR · 2021-11-02

**Correctness:** 3
**Technical Novelty And Significance:** 3
**Empirical Novelty And Significance:** 3
**Recommendation:** 6
**Confidence:** 3

**Main Review:**

strengths:

A new method to learn rotation matrix in approximate nearest neighbor search in proposed to replace the SVD based solution. The proposed iterative Givens rotation based method is applicable to end-to-end neural network training.

A family of Givens coordinate block descent algorithm are proposed to learn the rotation matrix.

Experimental results on both fixed embeddings and learnable embeddings validate the effectiveness of the proposed Givens rotation based method.

The paper is clearly organized and easy to follow.

weaknesses:

Since the authors claim that the Cayley transform based methods are inefficient compared to the proposed method. Evidence of complexity analysis and/or experimental running time should be shown to support the claim.

While the authors provide complexity analysis, the running time of each part can help the reader to understand and compare different components of the proposed method and also different methods.

Comparison with other baselines besides OPQ, e.g. ITQ, is expected in the fixed embedding experiment. It would be great if the authors can find other baselines beside the Cayley for Section 3.2.

How much is the contribution of the quantization distortion loss to the final loss in (1). Can you provide an ablation study of the weight besides 1/m?

minor:

the notation of a product quantization function is expected to be R^{mxn} to R^{mxk}

**Summary Of The Paper:**

This paper proposes to learn rotation matrix by Givens coordinate descent algorithms in the context of minimizing the quantization distortion for efficient storage. The proposed family of Givens coordinate descent algorithms are based on geometric intuitions of the special orthogonal group and are provably convergent on any convex objectives. The experiments show that the proposed algorithms are much time efficient and lead to performance improvement in an end-to-end training of embedding indexes.

**Summary Of The Review:**

This paper is overall well written and the contributions can be clearly recognized. The method part is clear but the experimental part is not solid as expected. I tend to accept this paper at this stage and am looking forward to the authors' feedback.

---

> ### Author Response · Authors · 2021-11-15
> **Point to Point Responses**
>
> Thank you very much for the highly constructive feedback.
>
> 1. Regarding the complexity analysis of Cayley transform, we briefly mentioned in the runtime analysis section that, since Cayley involves the inversion of a dense matrix of size n x n, which takes O(n^3) FLOPs per example and cannot be parallelized easily, the overall runtime is dominated by this matrix inversion operation (as well as its gradient computation, which involves a similar matrix inversion).
>
> 2. We kept the runtime section short due to space limitation, but will give a more explicit calculation of the overall runtime in the next revision.
>
> 3. To our understanding, ITQ represents a precursor to OPQ that imposes more strict restrictions on the space of admissible rotation matrices. Since the OPQ paper claims better results than ITQ and appears more general, we chose to skip the comparison with ITQ. We have also considered implementing the matrix exponential based methods (such as Pade approximation discussed in Lezcano-Casado & Martinez-Rubio. "Cheap Orthogonal Constraints in Neural Networks: A Simple Parametrization of the Orthogonal and Unitary Group". ICML 2019), but the computation cost seems quite high for an approximate solution (even in theory). Instead we chose to focus on exact rotation optimization methods, which include Cayley and OPQ.
>
> 4. In terms of actual term value, the quantization distortion loss is about 90% in the total loss in Equation (1). However, the two terms in Equation (1) might not be directly comparable, since the first term, retrieval loss is softmax cross entropy and the second term, quantization distortion loss is mean square error. In practice, we find there indeed is a tradeoff between the two terms: retrieval loss and quantization distortion loss. For example, if we use a very large weight on quantization distortion loss, then the distortion is smaller but the overall retrieval performance is worse. But in general, unless we use a very large or very small weight, the tradeoff is not very sensitive to the weight hyperparameter, thus we use a constant 1 in all our experiments. We agree that this might be an interesting direction to explore, and we will provide an ablation study in the next revision.
>
> minor:
>
> 1. The product quantization function in our paper has been defined in a slight different way, to hide the unnecessary details. It includes both the encoder function  $\psi: \mathbb{R}^n \xrightarrow{} (1,\cdots,K)^D$ and decoder function
> $\rho: (1,\cdots,K)^D \xrightarrow{}  \mathbb{R}^n$. Formally, $\phi(x) = \rho(\psi(x)) $, which is $\mathbb{R}^n \xrightarrow{} \mathbb{R}^n$ for each example. So it is correct in the paper.

---

### Official Review · Reviewer_rUSS · 2021-11-02

**Correctness:** 3
**Technical Novelty And Significance:** 3
**Empirical Novelty And Significance:** 2
**Recommendation:** 6
**Confidence:** 4

**Main Review:**

This paper proposes a block coordinate descent algorithm for rotation learning. The algorithm is based on Lemma 1 and Theorem 1. The rotation matrix on SO(n) is decomposed into diverse simple Givens rotation matrices. Then the optimized variable is converted into these Givens rotation matrices so that the rotation matrix is always on SO(3) and the projection is not required anymore. The authors also discuss how to select the coordinate, including random strategy, greedy strategy, and steepest strategy. Different from the existing work, it considers multiple Given rotations matrices in one step.

## Pros

1) It seems novel to consider multiple Givens rotations in one step. The motivation is clear due to proper lemmas.
2) This paper is well-organized and easy to follow.

## Cons
1)  As the authors claim that this paper firstly considers multiple Givens rotations in one step, why does it have to use $n/2$ rotations? Could the number of rotations change? Or equivalently, is $n/2$ theoretically supported? Related discussions may be needed. Furthermore, if $n$ can be regarded as a hyper-parameter, it would be better to conduct ablation experiments of it.
2) The experimental results about time are necessary since Cayley works well on Industrial Dataset, MovieLens, and Amazon Books. It is also important to empirically show the different efficiencies of GCD-R, GCD-G, and GCD-S. By the way, why is GCD-G missed in Table 1? Since the major improvement compared with Cayley is achieved by GCD-S, the experimental results may be dissatisfactory. There should be some competitive methods with one Givens rotation in each step.
3) As the authors claim that '*Since this is our main proposed algorithm*', it would be better to provide some convergence analysis about GCD-G rather than GCD-R.

(Minor)

1)  In the 3D vision, there are also some works to estimate the rotation matrix, which are listed below. It would be better to mention the related methods as well.
    -  On the continuity of rotation representations in neural networks
    - An analysis of SVD for deep rotation estimation
2)  There are some inappropriate formulations:
    - In Theorem 1, $:=$ may be inappropriate as it is not the formal *definition* of $\langle R, X^T \nabla\mathcal L(X) \rangle$ but a derivation. Meanwhile, it may be improper to call it a *theorem*. It is more like a proposition.
    - In Algorithm 1, it would be better to use "*Output*" instead of "*Ensure*" as "*Input*" is used in the first line.



**Summary Of The Paper:**

This paper proposes a block coordinate descent algorithm for rotation learning. The algorithm is based on Lemma 1 and Theorem 1. The rotation matrix on SO(n) is decomposed into diverse simple Givens rotation matrices. Then the optimized variable is converted into these Givens rotation matrices so that the rotation matrix is always on SO(3) and the projection is not required anymore. The authors also discuss how to select the coordinate, including random strategy, greedy strategy, and steepest strategy. Different from the existing work, it considers multiple Given rotations matrices in one step.

**Summary Of The Review:**

The paper is well-organized and easy to follow. The idea to use multiple Givens rotations seems novel.

However, there are some flaws in the technical and experimental parts. One is the lack of discussion of the number of rotations in one step. It seems the authors fail to provide convincing explanations. Another problem is the missing comparison of consuming time. The method that uses only one Givens rotation is also necessary in Section 3.

---

> ### Author Response · Authors · 2021-11-15
> **Point to Point Responses**
>
> We thank the reviewer for the valuable comments. Below are our point to point responses.
>
> 1. For a rotation matrix of dimension n (assumed even), the maximal number of 2d Givens  rotation families that jointly commute with one another is n / 2. This is a consequence of the fact that any maximal torus in SO(n) is of dimension n / 2.  Note that all maximal tori are isomorphic to one another via conjugation. A concrete example of a maximal torus is simply the set of 2x2-block diagonal matrices, where each block is a 2d rotation. We did not explicitly prove an optimality result for the number n / 2, but from the proof of the convergence theorem of the random Givens rotation case, it is not hard to see that the rate of convergence is actually proportional to the number of jointly commuting rotations chosen at each step. Thus it is beneficial to maximize this number. For greedy and steepest Givens rotation algorithms, there is expected to be diminishing benefit for adding more rotations at each step, but we do not have a proof since there is no symmetry we can exploit when the rotations are uniformly random. Your suggestion to conduct further ablation study regarding smaller n is indeed an interesting study on the exact convergence/speed tradeoff and we are very keen on following up with that. For larger n, the experiment on non-independent simultaneous rotations shows poor convergence when the joint commutativity condition is relaxed.
>
>
> 2. In practice, we did find Givens rotation based methods run faster than Cayley method (the ratio is about 543 batches/sec vs 752 batches/sec). But due to space limitation, we did not report this in the paper, as we thought that the complexity analysis in Section 2.4 might already be convincing enough, at least theoretically. But we will consider putting back the empirical comparison of running speed in the next revision, if all the received feedback reflects its necessity.
>
> As illustrated in the first experiment in Section 3.1, GCD-R and GCD-S can be regarded as the lower bound and upper bound of GCD methods, and the GCD-G’s performance is in between and closer to the GCD-S. Thus, due to space limits, we did not provide GCD-G results in Table 1. But as you can imagine, the GCD-G result is indeed between GCD-R and GCD-S. We did not expect that this removal would cause confusion. If it really does, we will consider putting the GCD-G’s result back in the next revision.
>
> As for the one update per iteration, it is actually what we did for the end-to-end experiment in Section 3.2, as for the end-to-end scenario, it is more reasonable and straightforward to let all model parameters update at the same pace. We will emphasize this in the next revision.
>
> 3. The convergence proof of GCD-R relies crucially on the uniform randomness assumption of the choice of rotation axes pairs. This assumption provides useful symmetry to simplify the analysis of coordinate descent, allowing local improvement to imply global improvement. Similar symmetries cannot be found in GCD-G or GCD-S since the choices of rotation axes are not uniform. Instead we argued in the proof appendix that GCD-R is strictly dominated by the other two, which thus have better convergence speed than GCD-R, at least in the convex setting.
>
>
> (Minor)
>
> 1. We thank the reviewer for pointing this out. They are interesting and related. We will cite them and make a discussion in the next revision.
>
> 2. Agree, will make the change.

---

> > ### Comment · Reviewer_rUSS · 2021-11-17
> > **Responses**
> >
> > Thanks for your kind responses. The responses have addressed most of my concerns. I would like to read the reviews from the other reviewers and then decide whether to update my score.
> >
> > 1. *For larger n, the experiment on non-independent simultaneous rotations shows poor convergence when the joint commutativity condition is relaxed.* It will be better to show it directly in experiments.
> > 2. I still suggest that the authors put the results back, at least to the appendix.
> >
> > 3. I agree that the theoretical analysis may be enough as the authors claim that "*GCD-R and GCD-S can be regarded as the lower bound and upper bound of GCD methods*". I suggest that this claim should also be directly highlighted (bold or italic) in the main paper to improve the readability.

---

> > > ### Author Response · Authors · 2021-11-23
> > > **Revision summary and responses**
> > >
> > > Thanks a lot for your followup responses. Below are our brief responses.
> > >
> > > 1. We did show the poor convergence in the experiment. Please check Figure 2(a), where the overlapping GCD-G and GCD-R stands for “non-independent simultaneous rotations” rotations. We have to use a shorter name in the legend, so we use “overlapping” instead of “non-independent simultaneous rotations”.
> > > 2. Agree, we have put the GCD-G result back to Table 1 in the paper, and due to space limit, we have to modify the table a little bit by removing some not very important numbers. Now we think it may look more clear and easy-to-read. We also put back the empirical comparison of the running time in GPU between GCD and Cayley method in Appendix B.
> > > 3. Agree, we have added the text to the main paper (last page, in the last paragraph, right before the conclusion.)

---

### Official Review · Reviewer_kyJu · 2021-11-09

**Correctness:** 3
**Technical Novelty And Significance:** 2
**Empirical Novelty And Significance:** 3
**Recommendation:** 6
**Confidence:** 3

**Main Review:**

Although I could read the paper and understand it, I think the representation can benefit from some improvements. In particular, the authors avoided adding some definitions that could have helped a wider range of audiences to understand the paper.

-  In the "REVISIT TRAINABLE PQ INDEX" subsection, the author mentioned the loss function is intractable. Hence, they try to solve that by fixing X (input) and iteratively learning R (rotation matrix). But they don't explain why the loss function is intractable.

- The preliminaries notation has been introduced in the 2.2 section but it would be better if it was at the beginning of the method section.

-  It is interesting that authors tried to use the lesser noticed approach in geometry to address one of the current problems in the applied world. But they didn't develop a new technique. Although, the context that they used this method seems novel the method itself is not considered novel.



**Summary Of The Paper:**

Current rotation learning methods are trying to minimize quantization distortion for fixed embeddings, which are not applicable to an end-to-end training scenario where embeddings are getting updated constantly. Therefore, this paper tries to address this issue to fully enable end-to-end training of Product Quantization (PQ) based embedding index with retrieval models, by using mathematical studies of the decomposition of orthogonal group. They proposed a family of block Givens coordinate descent algorithms to learn rotation matrices that are provably convergent on any convex objectives by leveraging geometric intuitions from Lie group theory. Authors claimed that their algorithms are much more parallelizable, reducing runtime by orders of magnitude on modern GPUs, and converge more stably according to experimental studies in comparison to the state-of-the-art SVD method.

Their main contributions can be summarized as follows:
- Changing the landscape of learning rotation matrix in approximate nearest neighbor (ANN) search from SVD based to iterative Givens rotation-based, to be applicable to end-to-end neural network training.

- Proposing a family of Givens coordinate block descent algorithm with complexity analysis and convergence proof.

- Proves that for the fixed embedding, their algorithm shows similar convergence result as the existing rotation matrix learning algorithms. Therefore, their proposed algorithm is able to learn the rotation matrix more effectively for the end-to-end training.

**Summary Of The Review:**

It was an interesting paper because of the way that authors bridged from geometry to solve an applied problem with neural networks. But it has some room for improvement as I mentioned in the detailed review. Although the approach is interesting the technique is not that novel.

---

> ### Author Response · Authors · 2021-11-15
> **Point to Point Responses**
>
> We thank the reviewer for the valuable comments. Below are our point to point responses.
>
> 1. We did shortly state in the text,
>
> “Note that straight-foward optimization of $\L$ is intractable, since the quantization function $\phi$ is not differentiable,..... Then the only remaining unaddressed problem is how to iteratively learn the rotation matrix $R$”.
>
> But we agree that this explanation might be too brief, thus confusing. We will try to clarify more in this subsection in the next revision.
>
> 2. Thanks for this suggestion, we will consider making a notation section at the beginning of the method section, since we did have a lot of notations defined everywhere.
>
> 3. We believe our technique is indeed novel as it combines insights from coordinate gradient descent and maximal tori from Lie theory, to produce a family of provably convergent convex optimization algorithms that reach or exceed state of the art methods such as those based on SVD or Cayley transforms, with various trade-offs between efficiency and empirical quality of convergence. An important distinction with conventional coordinate descent is that due to the manifold setting of SO(n), the coordinate directions change from one step to the next. Indeed we are exploiting the left-invariance of the Givens coordinate under matrix multiplication, which may not be familiar to a non-geometric audience. Another prominent distinction is the use of non-disjoint coordinate pairs at each step, which avoids undesired correlation between rotations within a single step, and allows convergence proof to go through; furthermore this is intuitively the best one could do without introducing messy interactions, whence the name of maximal tori.

---

### Decision · Program_Chairs · 2022-01-20

**Decision:**

Accept (Poster)

**Comment:**

The paper introduces a method to learn rotations of a quantized embedding end-to-end. The proposed technique seems novel, although the technical/algorithm novelty seems to be somewhat marginal.
The empirical results are promising, although do not quite match some of the claims by the authors.
Hopefully the reviewer feedback would help in producing an even more influential paper.